

# Aura-biomes are present in the water layer above coral reef benthic macro-organisms

Kevin Walsh[1], J. Matthew Haggerty[1], Michael P. Doane[1], John J. Hansen[1], Megan M. Morris[1], Ana Paula B. Moreira[2], Louisi de Oliveira[2], Luciana Leomil[3], Gizele D. Garcia[3,4], Fabiano Thompson[2] and Elizabeth A. Dinsdale[1]

[1] Department of Biology, San Diego State University, San Diego, CA, United States of America
[2] Instituto de Biologia, Universidade Federal do Rio de Janeiro, Rio de Janeiro, Brazil
[3] Macae campus, Federal University of Rio de Janeiro, Macae, Rio de Janeiro, Brazil
[4] Laboratory of Microbiology, Institute of Biology, Federal University of Rio de Janeiro (UFRJ), Rio de Janeiro, Brazil

Corresponding author
Kevin Walsh, kwalsh627@aol.com

## ABSTRACT

As coral reef habitats decline worldwide, some reefs are transitioning from coral- to algal-dominated benthos with the exact cause for this shift remaining elusive. Increases in the abundance of microbes in the water column has been correlated with an increase in coral disease and reduction in coral cover. Here we investigated how multiple reef organisms influence microbial communities in the surrounding water column. Our study consisted of a field assessment of microbial communities above replicate patches dominated by a single macro-organism. Metagenomes were constructed from 20 L of water above distinct macro-organisms, including (1) the coral *Mussismilia braziliensis*, (2) fleshy macroalgae (*Stypopodium*, *Dictota* and *Canistrocarpus*), (3) turf algae, and (4) the zoanthid *Palythoa caribaeorum* and were compared to the water microbes collected 3 m above the reef. Microbial genera and functional potential were annotated using MG-RAST and showed that the dominant benthic macro-organisms influence the taxa and functions of microbes in the water column surrounding them, developing a specific "aura-biome". The coral aura-biome reflected the open water column, and was associated with *Synechococcus* and functions suggesting oligotrophic growth, while the fleshy macroalgae aura-biome was associated with *Ruegeria*, *Pseudomonas,* and microbial functions suggesting low oxygen conditions. The turf algae aura-biome was associated with *Vibrio*, *Flavobacterium,* and functions suggesting pathogenic activity, while zoanthids were associated with *Alteromonas* and functions suggesting a stressful environment. Because each benthic organism has a distinct aura-biome, a change in benthic cover will change the microbial community of the water, which may lead to either the stimulation or suppression of the recruitment of benthic organisms.

## INTRODUCTION

Coral reef ecosystems are diverse but declining habitats (*Jackson & Buss, 1975*; *Hixon & Beets, 1993*; *Cantera et al., 2003*; *Hughes et al., 2010*). Causes of coral cover decline are

associated with overfishing, disease, increased nutrients, water runoff, and increased water temperatures (*Roberts & Nicholas, 1993*; *Hughes, 1994*; *Weil, Smith & Gil-Agudelo, 2006*; *Hoegh-Guldberg et al., 2007*; *Hughes et al., 2007*; *De'ath, Lough & Fabricius, 2009*). Microbial communities are also integral in coral reef health and stability. Microbes associated with the water of pristine coral reefs show a mix of autotrophs and heterotrophs, while the water of degraded reefs is dominated by microbial heterotrophs including many pathogenic strains (*Dinsdale et al., 2008a*; *Morrow et al., 2012*). The increase of pathogenic microbes in the reef water column is correlated with an increase in the amount of coral disease (*Dinsdale et al., 2008a*). There are several hypotheses explaining the increase of pathogens on coral reefs; the first is that microbes are transported from agricultural and human sewage runoff into the ocean, and the second is that the microbial changes are being generated on the reef (*Dinsdale & Rohwer, 2011*).

Early investigations of microbial associations with corals showed a strong relationship between the host macro-organism and their bacterial symbionts. *Rohwer et al. (2002)* showed that microbial communities of coral are host-specific, with the same species of coral sharing microbial communities over space and time compared with a different species that occupies an adjacent location on the same reef. The combination of the coral macro-organism, its symbiotic zooxanthellae and the associated microbial communities was termed the holobiont. This concept has been expanded to various reef macro-organisms including multiple species of macroalgae (*Harder et al., 2012*; *Egan et al., 2013*) and zoanthids (*Sun et al., 2014*), in addition to terrestrial organisms including plants, insects and humans (*Mandrioli & Manicardi, 2013*; *Minard, Mavingui & Moro, 2013*; *Meadow et al., 2015*; *Vandenkoornhuyse et al., 2015*). Besides retaining a distinct microbial community on their surface, reef benthic organisms may influence microbes in the surrounding water environment. This can happen in two ways, (1) by the host-specific microbes being released or shed from the host into the surrounding water, and (2) the production of dissolved organic matter by the host, which stimulates the activity of a specific set of microbes within the boundary water layer.

Benthic organisms, like any submerged object, cause a variation in water flow across their surface, creating sheer forces and boundary layers (*Shashar et al., 1996*). Three boundary layers, the benthic (BBL), momentum (MBL), and diffusive (DBL; the closest to the benthic organism surface) (*Barott & Rohwer, 2013*), act at different scales and will be influenced by the underlying benthos and the surrounding water to varying degrees. The flow speed of water across benthic surfaces will alter the thickness of the layers and affect the chemical and biological makeup of each parcel of water. High flow rates across a reef may homogenize the compounds in the benthic boundary layer; however, the water in the diffusive boundary layer may be stagnant and concentrate chemicals and the responding microbes. Reef macro-organisms exude a range of chemicals into the boundary layers, altering the biogeochemical nature of the water and the microbial metabolisms that occur in the layers (*Smith et al., 2006*; *Haas et al., 2011*). Benthic organisms on a coral reef, such as coral, algae, and crustose coralline algae, influence the amount of dissolved organic carbon (DOC), dissolved oxygen, and bacterial abundance in the surrounding water (*Haas et al., 2010*; *Haas et al., 2011*). Fleshy macroalgae and algal turfs release more DOC than coral and have higher microbial

growth in the surrounding water column (*Wild et al., 2010*; *Haas et al., 2011*; *Haas et al., 2013b*). Turf algae, dominated by filamentous cyanobacteria, produce more DOC than other reef macro-organisms and can generate nearly 80% of all DOC on a reef (*Brocke et al., 2015*).

The increase of DOC in the water column can lead to a reduction of certain microbial taxa and stimulation of other microbial groups (*Nelson et al., 2013*). An *in vitro* experiment showed that the water column, that was dominated by oligotrophic microbial genera *Synechococcus and Pelagibacter*, became dominated by the family Vibrionaceae and genera *Pseudoalteromonas*, *Aeromonas*, and *Flavobacterium* when exposed to exudates from algae (*Nelson et al., 2013*). In contrast, the phylum Planctomycetes and families Bacteriovoraceae, Erythrobacteraceae, Kordiimonadaceae, and Hyphomonadaceae were the dominant microbial taxa when exposed to exudates from coral (*Nelson et al., 2013*). *In situ* studies also demonstrate a correlation between reef macro-organism cover and microbial community taxa in the water column (*Dinsdale et al., 2008a*; *Kelly et al., 2014*; *Tout et al., 2014*). Microbial communities on coral-dominated reefs show a higher proportion of sequences similar to phyla Cyanobacteria and Firmicutes, and class Alphaproteobacteria. In contrast, reefs with high algae cover (up to 68%) and low coral cover have ten times the microbial abundance compared to a coral-dominated reef, with a higher proportion of sequences similar to phylum Bacteroidetes, classes Gammaproteobacteria and Betaproteobacteria, and opportunistic pathogens from the *Vibrio* genera (*Dinsdale et al., 2008a*; *Bruce et al., 2012*; *Kelly et al., 2014*). On a broader scale, the microbial communities above a healthy reef can be described as copiotrophic in general, whereas microbial communities adjacent to the reef above a sandy substrate, or in open water off the reef are described as more oligotrophic (*Tout et al., 2014*).

Microbial community characteristics vary across coral reefs and among sites within reefs, and these changes correlate with coral cover (*Kelly et al., 2014*; *Tout et al., 2014*). Because of the interactions between a benthic organism and the surrounding water layers, we hypothesized that within a single reef ($\leq 50$ m$^2$) there will be a mosaic pattern of microbial communities in the boundary layers surrounding each benthic organism. We propose that each benthic organism influences the microbiome in the water column boundary layers, promoting a benthic organism-specific microbial community that we call the 'aura-biome'. To test our aura-biome hypothesis, we take a shotgun metagenomics approach (*Dinsdale et al., 2008b*) and describe the microbial communities in the water column directly above and around (momentum and diffusive boundary layers) different benthic reef organisms. We tested multiple benthic organisms, including coral, fleshy macroalgae, turf algae, and zoanthids on a single reef of the Abrolhos Bank in the South Atlantic.

## METHODS

### Field site

We conducted the study on the Abrolhos Bank coral reefs, which are situated on a 45,000 km$^2$ expansion of the continental shelf in the southern Bahia state of Brazil. The Abrolhos Bank supports the largest coral reefs in the South Atlantic, but coral cover has declined over the last decade (*Francini-Filho et al., 2008*; *Francini-Filho et al., 2013*). We

focused our study on the reefs surrounding Ilha Santa Bárbara ($-18.033333$, $-38.6668038$), which is in a marine protected area. The island is about 60 km from the coast with no agricultural runoff, and housing for about 10 people. The reef site was approximately 50 m$^2$, within which we sampled the water column above replicate patches of each dominant organism. The sampling was conducted over two days (23 and 24 June 2014). The first dive was in the afternoon of 23rd June, and the second and third dives were in the morning and afternoon of 24th June. Three separate sampling dives were taken to allow time to filter the water for microbial collection. The research was conducted under a federal government license (SISBIO no. 10112 - 2). We received this license to access protected areas from Parque Nacional Marinho de Abrolhos/IBAMA (Instituto Brasileiro do Meio Ambiente e dos Recursos Naturais Renova'veis).

Collection of water was conducted on small patches of the reef where a single macro-organism dominated the benthos. A 1 m$^2$ quadrat was placed haphazardly on selected patches and multiple photographs were taken of each sampling plot. Care was taken when placing the quadrat to not cause excessive water movement and disturbance the boundary layers. Photographs were taken of each quadrat and 30 points within the plot were measured to determine the percent cover of the benthic organism in each plot (Fig. 1). A Manta2 Series Multiprobe$^{TM}$ handheld data logging instrument was placed on the surface of the benthic organism to log water physio-chemistry on each dive (Fig. S1).

## Microbial sampling technique

Microbial samples were collected using a bilge pump and bag unit. Approximately 20 L of water was pumped directly off the benthic surface (the pump was held about 1 cm above the surface of the organism, but no further than 5 cm above the surface). The end of the bilge was placed 1–5 cm directly above the substrate, making sure the pump did not touch the reef macro-organism. The water collected was a combination of the diffusive and momentum boundary layers, as described in *Barott & Rohwer (2013)*, and makes up the aura-biome. In addition, 4 cm above a coral was identified as the location with the highest abundance of heterotrophic bacteria (*Seymour et al., 2005*), and this local maxima was targeted by the sampling. The water surrounding four distinct macro-organisms was collected for metagenome construction. The four reef macro-organisms included; (1) coral (*Mussismilia braziliensis*) ($n = 4$), (2) fleshy macroalgae, characterized by numerous genera, including *Stypopodium*, *Dictyota* and *Canistrocarpus* ($n = 3$), (3) benthic turf algae, characterized by closely cropped long red filaments of both red algae and cyanobacterial mats ($n = 3$), (4) zoanthid (*Palythoa caribaeorum*) ($n = 2$), and we also collected water column samples (facing the open ocean $\sim$3 m above the reef) ($n = 4$). The four reef organisms and water column samples are our treatments for the statistical analysis and the number of replicate organisms is given in parenthesis and the total number of metagenomes constructed was 16.

Water collected above each macro-organism was transferred to niskin bottles using a 20 $\mu$m filter to remove larger organisms, such as diatoms and phytoplankton. The microbes collected from all 20 liters were pressure driven by compressed air through a 0.22 $\mu$m sterivex filter. Water from all samples was collected in duplicate sterivexes, except the turf

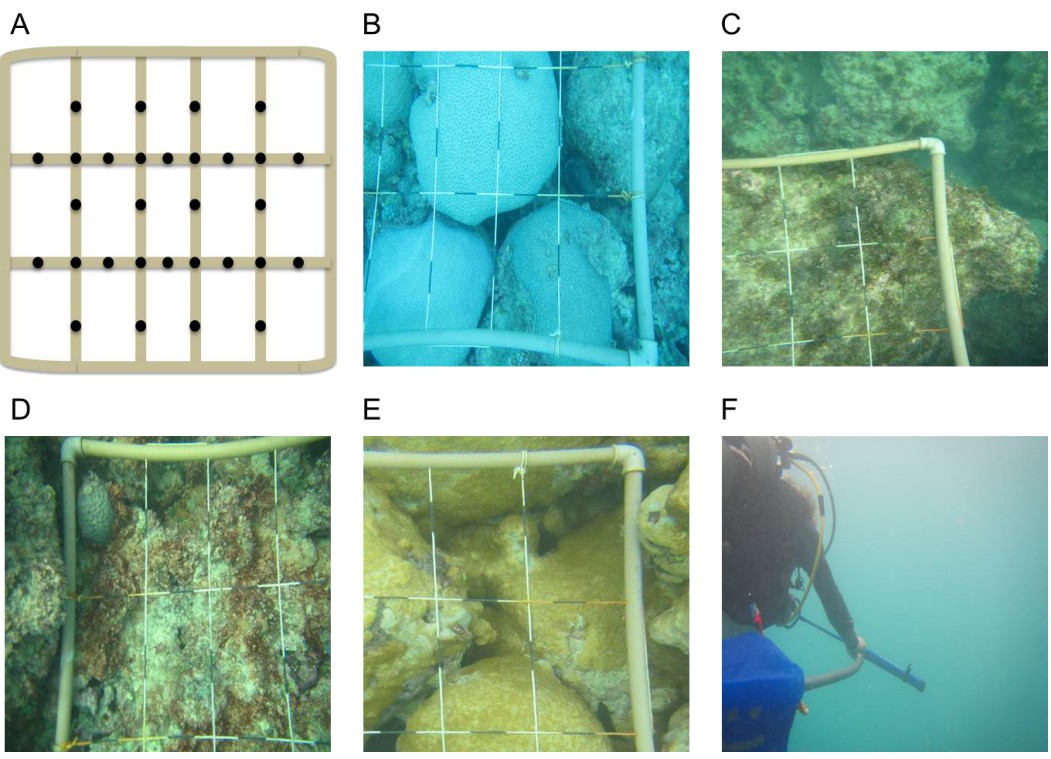

**Figure 1** **Percent cover of benthic organisms.** Multiple photographs were taken to accurately describe percent cover of macro-organism within each quadrat. (A) Points were selected on a 1 m × 1 m quadrat. A representative photograph of each macro-organism tested including, (B) *M. braziliensis*, (C) Fleshy macro-algae, (D) Turf algae, (E) *P. caribaeorum*, and (F) Water Column sampling.

algae water samples, which were each collected on a single sterivex. Sterivexes were wrapped with parafilm, placed in a ziploc bag, and stored in liquid nitrogen until extraction of DNA.

## DNA extraction and sequencing

Microbial DNA was extracted from sterivex filters with lysis buffer and proteinase K (*Bruce et al., 2012*) and purified using the Nucleospin Tissue column purification protocol (Macherey Nagel, Düren, Germany). Extracted and purified DNA was quantified using a Qubit (Life Technologies, Carlsbad, CA, USA) to ensure that each sample contains the minimum amount of DNA required for sequencing. Shotgun metagenomics, a proven technique used to describe microbial communities (*Dinsdale et al., 2008a*; *Dinsdale et al., 2008b*; *Hugenholtz & Tyson, 2008*; *Kelly et al., 2014*; *Haggerty & Dinsdale, 2017*), was used to explain the taxonomy and functional pathways present in the microbes from each parcel of water. Shotgun metagenomes are sequenced without amplification or primers, such that a random selection of the microbial DNA including all gene areas are sequenced, and identified using bioinformatics techniques (described below). Following Illumina protocols, metagenomic libraries were created from each sample using 100 ng of starting input DNA using the TruSeq DNA PCR-Free Library Preparation Kit. Libraries were paired-end sequenced on an Illumina MiSeq with a v3 600-cycle reagent kit.

## Metagenomic analysis

Metagenomic sequences were run through the bioinformatics tool PRINSEQ to remove and trim any low quality sequences, including exact duplicates, those that contained N's, and sequences that had a Q-score of less than 20 (*Schmieder & Edwards, 2011*). Sequences were uploaded to the Metagenomics Rapid Annotation Server (MG-RAST) for taxonomic and functional annotation (*Meyer et al., 2008*), using the minimum cutoff parameters of $1 \times 10^{-5}$ *e*-value (*Bruce et al., 2012*; *Garcia et al., 2013*), 70% identity, and alignment length of 30 nucleotides. These parameters are identified as providing a conservative estimate of both the taxa and function. MG-RAST compares the sequences from the metagenome to the database to identify the best hit classification within the database (*Meyer et al., 2008*). Taxonomic classifications used the SEED database as a reference, while functional classifications used SEED's Subsystem Annotation. The SEED annotation describes metabolic processes in a hierarchical scheme (*Overbeek et al., 2005*).

## Statistical analysis of the metagenomes

The microbial communities were described by comparing the proportion of sequences that matched each microbial organism in each metagenome. First, we described the proportion of sequences at the domain level. Second, we described the genera present in each metagenome. Third, we described the proportion of sequences in the most abundant 20 genera that vary across the treatments, or aura-biomes. The functions in the microbial community were compared by investigating the proportion of sequences similar to each metabolic group. The SEED follows a hierarchical scheme, which includes broad metabolic groups, such as carbohydrate metabolism, and these groups are broken down into specific subsystems, for example carbon monoxide dehydrogenase. We tested whether the proportion of sequences in each genera or metabolism varied between aura-biomes using an analysis of variance (ANOVA) with a post-hoc Tukey test. Statistical analysis was conducted on the Statistical Analysis of Metagenomic Profiles (STAMP) package (*Parks & Beiko, 2010*).

To visualize whether the microbial community above each macro-organism had a distinguishing taxonomic or functional profile, two canonical discriminant analyses (CDA) were conducted using SPSS, similar to techniques described in *Dinsdale et al. (2013)*. CDAs use linear correlations of variables, in this case taxa or function, that drives the differences within treatments (*Dinsdale et al., 2013*). The position of each metagenome reflects the frequency combination of sequences associated with each variable; the vectors indicate which variable determines the distribution of metagenomes. Metabolisms that showed a statistical difference between treatments were explored further by comparing differences in the proportion of sequences in each gene pathway using an ANOVA with a post-hoc Tukey test within STAMP.

## RESULTS

Four groups of benthic macro-organisms, coral ($n = 4$), fleshy macroalgae ($n = 3$), turf algae ($n = 3$), and zoanthid ($n = 2$) were surveyed at the Ilha Santa Barbára reef site ($\leq 50$ m$^2$) for percent benthic cover, surrounding water physiochemical properties, and boundary

layer microbial community structure (Fig. 1, Table S1). We also included a fifth treatment group with samples collected from the open water column off the reef (~3 m above the reef $n_{total} = 16$). Mean benthic percent cover was calculated from quadrat images taken for each replicated macro-organism group. For the four groups, coral had a mean cover of 78.3 $\pm$ 2.9%, fleshy macroalgae had a mean cover of 93.8 $\pm$ 2.3%, turf algae had a mean cover of 83.3 $\pm$ 3.3%, and zoanthid had a mean cover of 88.3 $\pm$ 5.0% on the benthos. During the three dives, the water was characterized by a mean temperature of 25.36 $\pm$ 0.09 °C, pH of 8.18 $\pm$ 0.03, chlorophyll concentration of 2.96 $\pm$ 0.44 (μg/l), and dissolved oxygen of 107.22 $\pm$ 3.56% saturation and 7.09 $\pm$ 0.23 mg/l (Fig. S1).

Constructed metagenomes averaged 1,294,121 sequences per metagenome with each sequence having an average length of 277 $\pm$ 82 bp (Table S2). Bacteria was the most abundant domain in the metagenomes, averaging 92.5 $\pm$ 0.5% of sequences across all samples (Fig. S2), and was not significantly different in any treatment ($p = 0.053$). In the metagenomes collected above zoanthid, domain Eukaryota was significantly higher than in the metagenomes collected above turf algae ($p < 0.05$). Viruses were more abundant in the metagenomes from the water column, coral, and turf algae treatments ($p < 0.05$) compared with the other two treatments. Because the total abundance of Eukaryota and viruses in the metagenomes accounted for only 3.16% of the annotated sequences, we did not investigate further, but instead focused the analysis on the domain Bacteria. Within Bacteria, sequences matched to the phylum Proteobacteria were six times higher (63.4 $\pm$ 6.7% of the metagenomes) than the second most abundant phylum, Cyanobacteria (10.7 $\pm$ 3.1%), and third most abundant phylum, Bacteroidetes (10.0 $\pm$ 1.5%) (Fig. S3).

Of the 347 bacterial genera found across all treatments (Table S3), the twenty most abundant genera represented 59.5% of all sequences and were investigated further (Fig. 2). *Candidatus Pelagibacter* was the most abundant genus within all samples and was the most abundant genus in the coral aura-biome (17.2 $\pm$ 5.8%) and the water column (14.3 $\pm$ 4.2%) treatments. *Alteromonas* was the most abundant genus in the zoanthid aura-biome (27.2 $\pm$ 8.9%). *Synechococcus* had high proportions of sequences in both the coral aura-biome and water column microbiome (14.3 $\pm$ 5.9%). *Vibrio* was the most abundant genus in the turf algae aura-biome (11.0 $\pm$ 8.8%). Of the top twenty genera, only *Alteromonas* was significantly different between treatments (ANOVA $F_{df=4} = 4.761$ $p = 0.018$). The proportion of sequences of *Alteromonas* was significantly higher in the zoanthid aura-biome compared with the fleshy macroalgae aura-biome (Tukey $p = 0.020$), turf algae aura-biome (Tukey $p = 0.015$), and water column (Tukey $p = 0.046$).

A CDA conducted on the twenty most abundant genera showed the metagenomes group together based on the macro-organism they were collected above and axis 1 and 2 explained 81.2% of the variance between treatments (Fig. 3). The separation of the turf algae aura-biome is driven by abundance of genera *Vibrio* and *Flavobacterium*, while zoanthid was driven by the abundance of *Alteromonas*. The separation of the fleshy macroalgae aura-biome was driven by the abundance of genera *Ruegeria* and *Roseovarius*, while coral was influenced by the abundance of *Synechococcus*. The water column appeared to be the most dissimilar treatment compared to the four macro-organism treatments, and was driven by many genera including *Synechococcus*, *Shewanella*, and *Leeuwenhoekiella*.

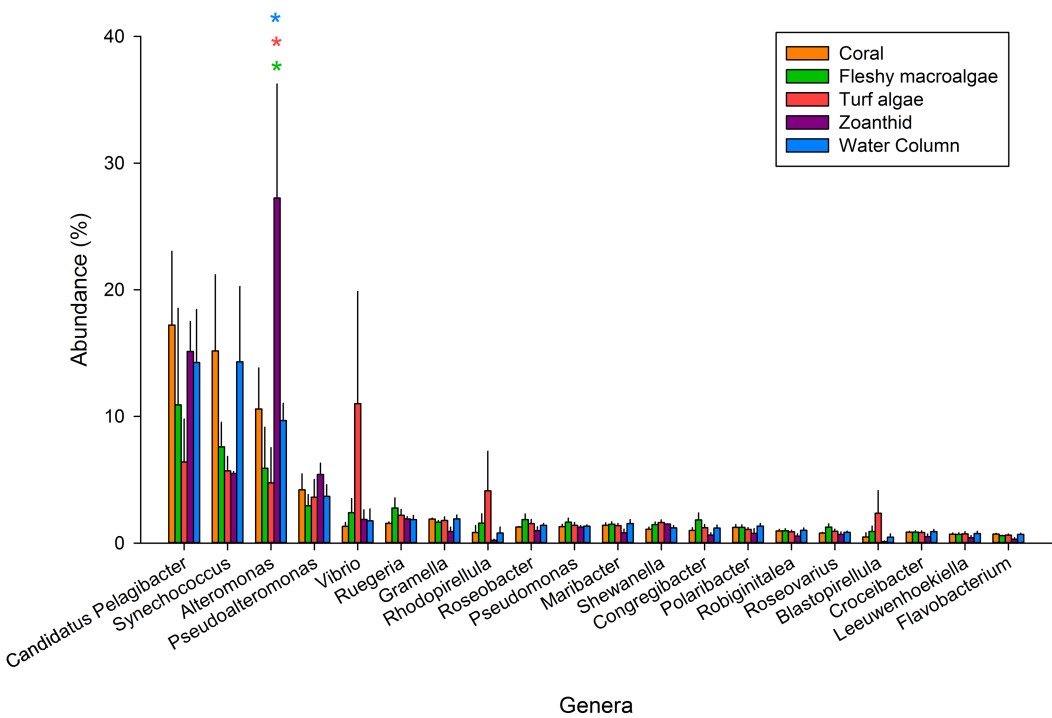

**Figure 2** **The 20 most abundant genera across the five samples.** An asterisk above each genera shows significance differences, while color delineates which samples varied.

Microbial metagenomes were compared for their metabolic annotation at broad functional levels down to specific pathways. At the broadest level there were 27 functional subsystems, with the most abundant metabolic pathways across all treatments being amino acids and derivatives ($12.5 \pm 0.1\%$), carbohydrates ($12.4 \pm 0.1\%$), protein metabolism ($7.3 \pm 0.1\%$), and cofactors, vitamins, prosthetic groups and pigments ($6.4 \pm 0.1\%$) (Fig. 4). Of the 27 broad metabolisms, five were significantly different between aura-biomes. Membrane transport (ANOVA $F_{df=4} = 3.618$, $p = 0.041$) was significantly higher in zoanthid aura-biomes compared with the coral aura-biome (Tukey $p = 0.045$) and the water column (Tukey $p = 0.035$). The zoanthid aura-biome had a significantly higher abundance of genes within the phages, prophages, transposable elements, and plasmids subsystem (ANOVA $F_{df=4} = 6.636$, $p = 0.006$) compared with all other treatments (Tukey macroalgae $p = 0.046$, coral $p = 0.004$, turf algae $p = 0.014$, water column $p = 0.007$). The fleshy macroalgae aura-biomes had a significantly lower abundance of genes within phosphorus metabolism (ANOVA $F_{df=4} = 7.276$, $p = 0.004$) compared with the water column (Tukey $p = 0.022$), and aura-biomes of zoanthid (Tukey $p = 0.006$) and turf algae (Tukey $p = 0.006$). The zoanthid aura-biomes were significantly lower in protein metabolism genes (ANOVA $F_{df=4} = 4.234p = 0.026$) compared to the water column (Tukey $p = 0.021$). The respiration pathway (ANOVA $F_{df=4} = 5.617$, $p = 0.010$) varied between treatments with zoanthid (Tukey $p = 0.015$) and fleshy macroalgae (Tukey $p = 0.027$) aura-biomes have significantly higher genes compared with the turf algae aura-biomes.

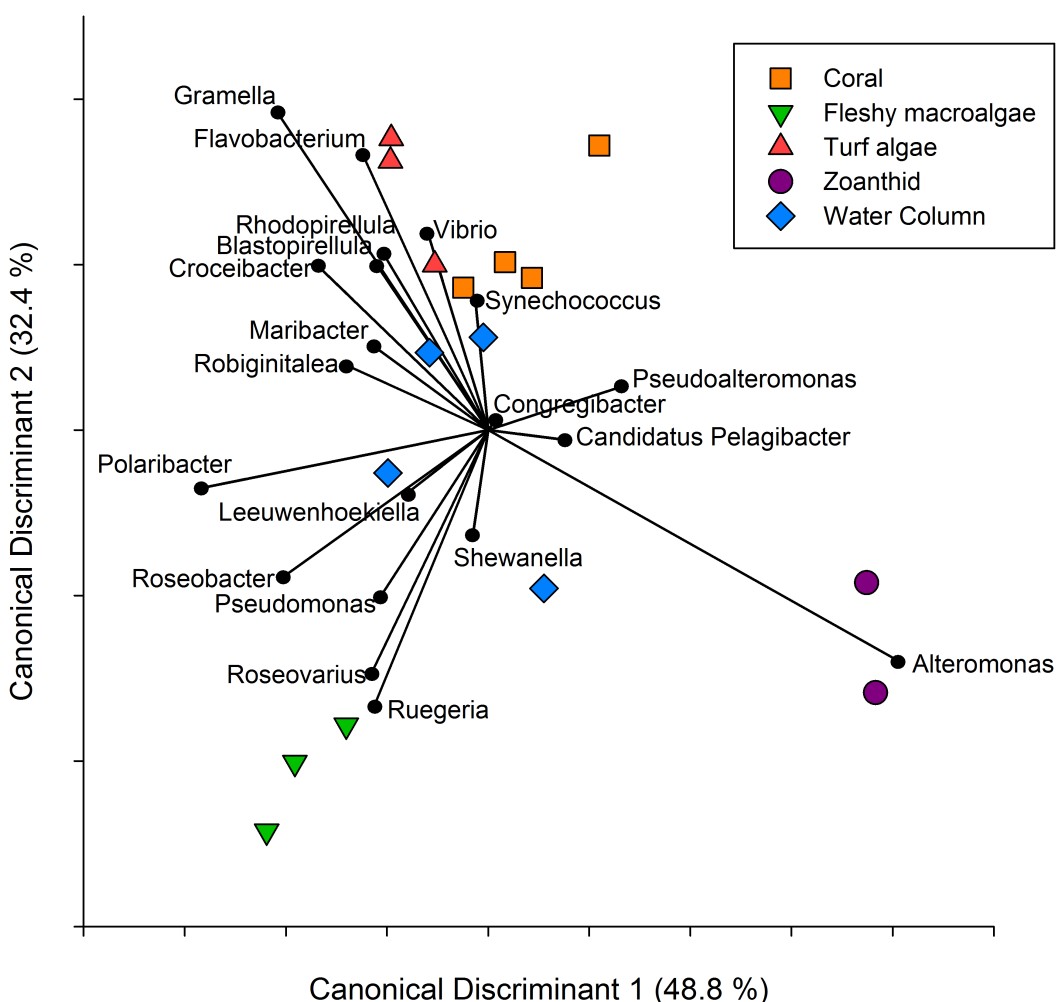

**Figure 3** **Canonical Discriminant Analysis based on taxa.** A CDA was run using the genus level of the aura-biomes or microbial communities to determine which genera drove differences between groups.

Similar to the taxonomic analysis, a CDA was constructed for the metabolic analysis using the proportion of metagenomic sequences annotated to the 27 broad functional subsystems. The CDA showed that metagenomes collected above each macro-organism group together by treatment, and the two axes explained 93.7% of the variance between treatments (Fig. 5). Metabolisms including cell division and cell cycle, dormancy and sporulation, motility and chemotaxis, and cofactors, vitamins, prosthetic groups and pigments were overrepresented in the coral aura-biome. Potassium metabolism was overrepresented in the fleshy macroalgae aura-biome. The sulfur, phosphorus, secondary metabolism, and virulence, disease and defense were overrepresented in the turf algae aura-biome. Stress response, respiration, and membrane transport metabolisms were overrepresented in the zoanthid aura-biome. Nucleosides and nucleotides, and regulation and cell signaling were overrepresented in the water column metagenomes. Each of these broad pathways contained more specific functions that varied by treatment, which was analyzed further (Table 1).

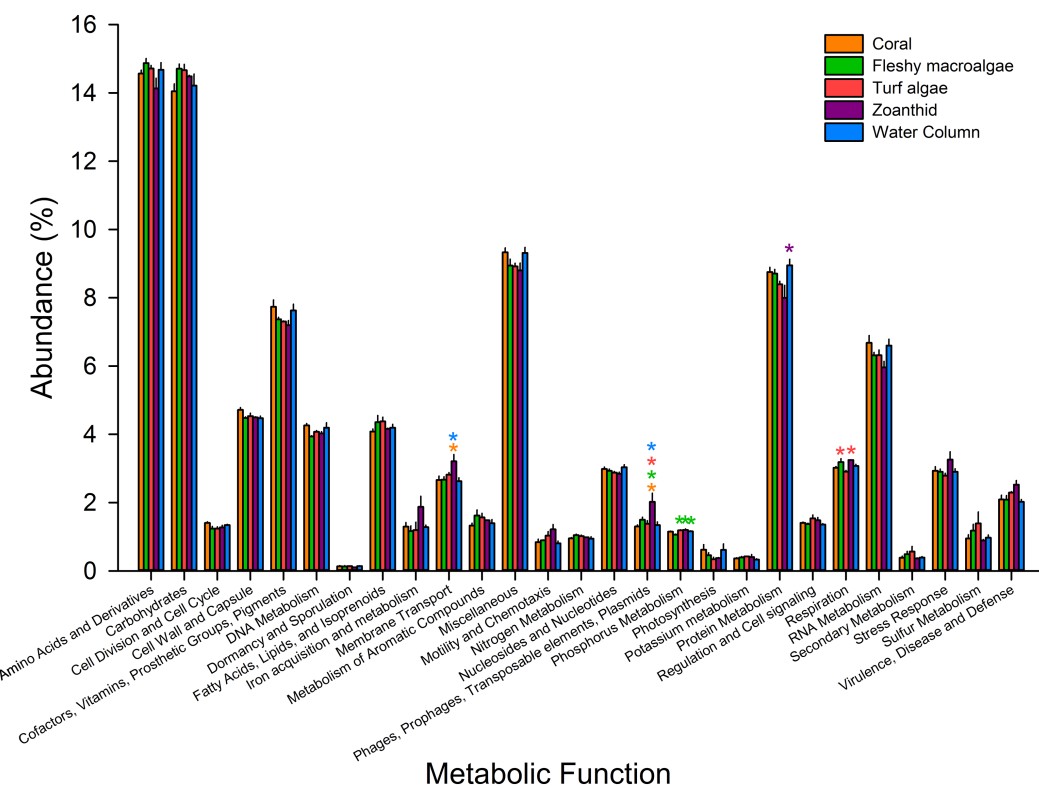

**Figure 4** **Metabolic pathways at the most broad level.** Pathways were averaged between treatments and those that differed significantly between treatments were visualized. Asterisks above each sample signify significance, while color of asterisk delineates which treatment it is greater than.

The coral aura-biome had seven specific metabolisms that were overrepresented within six broad level functions. Six specific pathways overrepresented in the coral aura-biome were also overrepresented in the water column samples, and included methicillin resistance, DNA repair base excision, Pterin metabolism 3, Riboflavin to FAD and YgfZ. *De Novo* purine biosynthesis, and Mnm5U34 biosynthesis. Pathways that were exclusively overrepresented in the water column included YgjD and YeaZ, and tRNA.

The fleshy macroalgae aura-biome had two pathways associated with respiration that were overrepresented - carbon monoxide dehydrogenase maturation factors and methanogenesis strays (Table 1). The turf algae aura-biome had another 10 pathways that were overrepresented, eight of which were within the membrane transport pathway, one within the respiration pathway, and one within the nucleoside and nucleotide pathway. These included NhaA, NhaD and sodium-dependent phosphate transporters; fructose and mannose inducible PTS; galactose-inducible PTS; sucrose-specific PTS; phosphoglycerate transport system; type III secretion; pyrimidine conversions; and tetrathionate respiration. Finally, Fap amyloid fiber secretion and general secretion were overrepresented in both the turf algae and zoanthid aura-biomes.

The zoanthid aura-biome had 16 specific pathways that were overrepresented in nine broad metabolisms (Table 1). The specific pathways overrepresented include Phd-Doc,

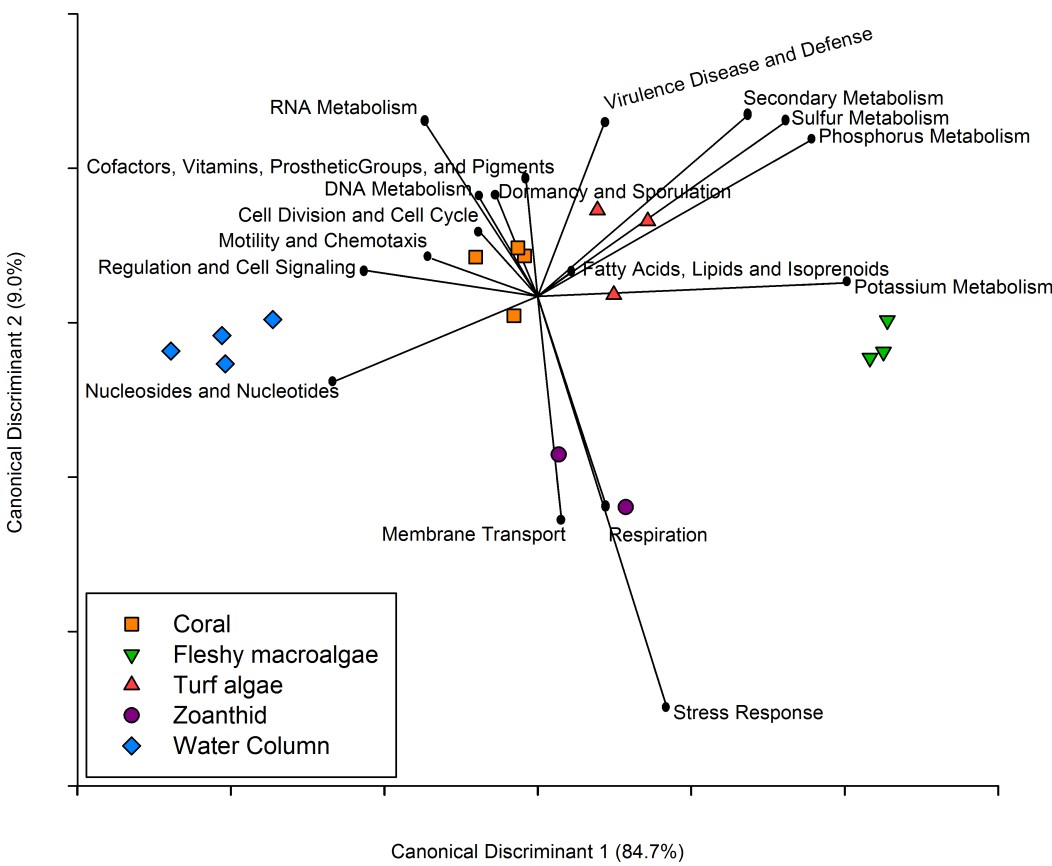

**Figure 5** **Canonical Discriminant Analysis based on metabolic functions.** The metabolic pathways that drove differences between each of the four macro-organisms and water samples.

YdcE-YdcD toxin/antitoxin, purine utilization, bacterial hemoglobins, universal stress protein family, respiratory dehydrogenases I, terminal cytochrome C oxidases, adhesions, arsenic resistance, phosphate uptake, phosphate-binding DING proteins, group II intron-associated genes, polyadenylation specificity factors, RNA polymerase III initiation factor, and 2-phosphoglycolate salvage.

## DISCUSSION

Water column microbiomes are correlated with cover of benthic organisms between reefs (*Dinsdale et al., 2008a*; *Kelly et al., 2014*; *Haas et al., 2016*; *Tout et al., 2014*). Here we show that the relationship between microbes and benthic macro-organisms occurs within a single reef, where the boundary layer aura-biome, defined here as the water directly above and around a macro-organism, follows a mosaic pattern of the dominant benthic organism. Coral cover in the Abrolhos Island reefs varies from 3–39% (*Oigman-pszczol & Creed, 2004*; *Leao et al., 2010*; *Francini-Filho et al., 2013*). We exploited the variations in benthic cover to show that the microbial community in the water above replicate patches varied depending on the dominant benthic organism.
**Table 1  Summary of significantly different metabolic processes.** These specific metabolisms were significantly over represented and found within the pathways driving differences between treatments in the metabolism CDA.

| Organism over represented | Broad metabolic processes | Gene pathways | ETA/$P$ value |
|---|---|---|---|
| Zoanthid | Regulation and cell signaling | Phd-Doc, YdcE-YdcD toxin-antitoxin | $\eta^2 = 0.662\ p = 0.012$ |
| Zoanthid | Nucleoside and nucleotides | Purine utilization | $\eta^2 = 0.650\ p = 0.014$ |
| Zoanthid | Stress response | Bacterial hemoglobins | $\eta^2 = 0.621\ p = 0.021$ |
| Zoanthid | Stress response | Universal stress protein family | $\eta^2 = 0.650\ p = 0.014$ |
| Zoanthid | Respiration | Respiratory dehydrogenases | $\eta^2 = 0.682\ p = 0.008$ |
| Zoanthid | Respiration | Terminal cytochrome C oxidases | $\eta^2 = 0.553\ p = 0.048$ |
| Zoanthid | Virulence disease and defense | Adhesions in staphylococcus | $\eta^2 = 0.658\ p = 0.013$ |
| Zoanthid | Virulence disease and defense | Arsenic resistance | $\eta^2 = 0.559\ p = 0.045$ |
| Zoanthid | Phosphorus metabolism | P uptake | $\eta^2 = 0.750\ p = 0.002$ |
| Zoanthid | Phosphorus metabolism | Phosphate-binding DING proteins | $\eta^2 = 0.713\ p = 0.005$ |
| Zoanthid | RNA metabolism | Group II intron-associated genes | $\eta^2 = 0.598\ p = 0.029$ |
| Zoanthid | RNA metabolism | Polyadenylation specificity factors | $\eta^2 = 0.697\ p = 0.006$ |
| Zoanthid | RNA metabolism | RNA polymerase III initiation factor | $\eta^2 = 0.562\ p = 0.044$ |
| Zoanthid | DNA metabolism pathways | 2-phosphoglycolate salvage | $\eta^2 = 0.670\ p = 0.011$ |
| Turf algae / Zoanthid | Membrane transport | Fap amyloid fiber secretion | $\eta^2 = 0.606\ p < 0.001$ |
| Turf algae / Zoanthid | Membrane transport | General secretion | $\eta^2 = 0.667\ p = 0.011$ |
| Turf algae | Membrane transport | NhaA, NhaD and Sodium-dependent phosphate transporters | $\eta^2 = 0.605\ p = 0.026$ |
| Turf algae | Membrane transport | Fructose and mannose inducible PTS | $\eta^2 = 0.578\ p = 0.036$ |
| Turf algae | Membrane transport | Galactose-inducible PTS | $\eta^2 = 0.623\ p = 0.021$ |
| Turf algae | Membrane transport | Sucrose-specific PTS | $\eta^2 = 0.611\ p = 0.024$ |
| Turf algae | Membrane transport | Phosphoglycerate transport | $\eta^2 = 0.835\ p < 0.001$ |
| Turf algae | Membrane transport | Type III secretion | $\eta^2 = 0.628\ p = 0.019$ |
| Turf algae | Nucleoside and nucleotides | Pyrimidine conversions | $\eta^2 = 0.559\ p = 0.045$ |
| Turf algae | Respiration | Tetrathionate respiration | $\eta^2 = 0.635\ p = 0.018$ |
| Turf algae / Fleshy macroalgae | Respiration | Methanogenesis strays | $\eta^2 = 0.800\ p = 0.026$ |
| Fleshy macroalgae | Respiration | Carbon monoxide dehydrogenase maturation factors | $\eta^2 = 0.557\ p = 0.046$ |
| Water Column | Cell division and cell cycle | YgjD and YeaZ | $\eta^2 = 0.702\ p = 0.006$ |
| Water Column | RNA metabolism | tRNA modification | $\eta^2 = 0.582\ p = 0.035$ |
| Coral / Water Column | RNA metabolism | Mnm5U34 biosynthesis | $\eta^2 = 0.566\ p = 0.041$ |
| Coral / Water Column | Nucleoside and nucleotides | De Novo purine biosynthesis | $\eta^2 = 0.648\ p = 0.015$ |
| Coral | Virulence disease and defense | Methicillin resistance | $\eta^2 = 0.635\ p = 0.017$ |
| Coral | DNA metabolism pathways | DNA repair base excision | $\eta^2 = 0.589\ p = 0.032$ |
| Coral | Cofactors, vitamins, prosthetic groups, and pigments | Pterin metabolism 3 | $\eta^2 = 0.626\ p = 0.020$ |
| Coral | Vitamins, prosthetic groups, and pigments | Riboflavin to FAD | $\eta^2 = 0.587\ p = 0.033$ |
| Coral | Vitamins, prosthetic groups, and pigments | YgfZ | $\eta^2 = 0.638\ p = 0.017$ |

The microbial communities present in the boundary layer above the host organism reflect a combination of environmental parameters, including the nutrients in the water column, the chemicals and microbes released by macro-organisms, as well as the abundance of predators and the water dynamics (i.e., tides, currents, and waves) (*Haas et al., 2011*; *Garren & Azam, 2012*). The genera and metabolic pathways present in the microbial communities in each of the aura-biomes provides insight into the micro-environment that is developing around each macro-organism (Fig. 6).

The functional repertoire in microbes above coral, including Riboflavin RNA processing and folate and pterines, were functions that suggest oxygenating growth; while the fleshy macroalgae aura-biome had enrichment in functions suggesting anoxic growth, such as
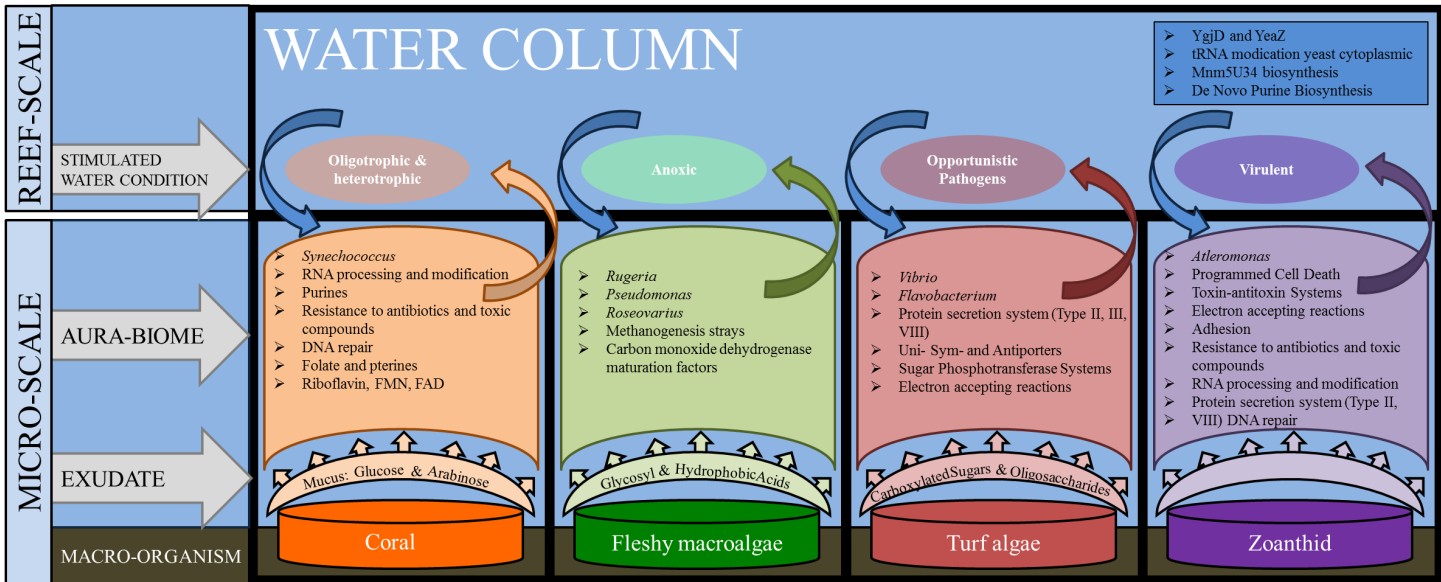

**Figure 6** **Interactions between macro-organisms and microbial communities on a reef space.** The exudate from the macro-organism induces and selects for communities whose taxa and metabolic functions create water conditions that may be detrimental to neighboring species (both macro and microbial).

methanogenesis and carbon monoxide dehydrogenase pathways, which are often found in anaerobic bacteria (*Thauer, 1998*). Previous studies have shown that microbial oxygen consumption differs between exposure to exudates from algae versus exudates from coral (*Haas et al., 2013b*). In an incubation study conducted on exudates derived from fleshy algae, microbial communities were stimulated and able to drawdown the dissolved organic carbon, whereas the coral exudate increased the dissolved organic carbon levels during the day (*Haas et al., 2013b*). Our results suggest that small patches of benthic fleshy macroalgae are creating anaerobic conditions within their aura-biome. While hypoxia was not measured in this experiment, direct measurements conducted in *Haas et al. (2013a)* documented lower oxygen rates above fleshy macroalgae. Together, these findings suggest that the fleshy macroalgae aura-biome has different oxygen content versus the coral aura-biome or open water column.

Turf algae, particularly those with high abundances of cyanobacteria, release high amounts of dissolved organic carbon (*Brocke et al., 2015*). In our experiment, the metagenomes constructed from the turf algae aura-biome, included two specific heterotrophic bacterial genera, *Vibrio* and *Flavobacterium*, suggesting a high amount of organic carbon is being released by the turf algae. The *Flavobacterium* genera includes bacterial pathogens known to cause disease in trout (*Crump et al., 2001*), while many species of *Vibrio* are well-known pathogens associated with declines in coral health, coral bleaching, and diseases (*Kushmaro et al., 2001*; *Cervino et al., 2004*; *Rosenberg & Falkovitz, 2004*; *Cervino et al., 2008*). In our study, one of the metagenomes above the turf algae had 28.7% of sequences showing similarity to *Vibrio*. Other studies found that *Vibrio* made up

30–60% of a cultured microbiome of diseased coral (*Ritchie et al., 1994*; *McGrath & Smith, 1999*) and up to 80% of taxa from metagenomes constructed from stressed corals (*Thurber et al., 2009*).

In addition to the increase in proportions of heterotrophic taxa in the turf algae metagenomes, a higher proportion of sequences were associated with carbohydrate metabolisms The presence of the carbohydrate metabolism suggests that these microbes were rapidly consuming the large amounts of dissolved organic carbon, which is being secreted into the water by the turf algae (*Brocke et al., 2015*). The increase of type II, III and IV secretion systems suggest that the microbes had functions that are often used in cell-to-cell and host-microbe interactions (*Christie, 2001*; *Alfano & Collmer, 2004*; *Cianciotto, 2005*). Given the high proportion of heterotrophs and potential pathogens and the increase of secretion systems, the aura-biome produced by the turf algae may be detrimental to the health of adjacent organisms.

Zoanthids had the most distinctive aura-biome compared to the other treatments. Previous researchers have measured lower DOC production rates from zoanthids compared to algae and coral (*Silveira et al., 2015*). Therefore, we suggest that unlike the pattern observed with turf algae, where high DOC production drives a shift in the aura-biome composition, other exudates from the macro-organism may be influencing the observed changes in the microbial community above the zoanthid.

The zoanthid aura-biomes were enriched in *Pseudoalteromonas* and *Alteromonas,* which have been negatively associated with coral cover (*Kelly et al., 2014*). The *Pseudoalteromonas* genus includes potential coral pathogens (*Ritchie, 2006*), and some *Alteromonas* species are associated with coral yellow band disease (*Sweet, Bythell & Nugues, 2013*). The functions induced in the zoanthid aura-biome included type II and VIII secretion systems, toxin/antitoxin system, resistance to antibiotics and toxic compounds, and DNA repair (Fig. 6). Type II and VIII secretion systems are often found in pathogenic microbes (*Olsén, Jonsson & Normark, 1989*; *Collinson et al., 1991*; *Sandkvist, 2001*). Toxin/antitoxin gene pathways are a response to stress, with YdcE as the toxin and YdcD as the inhibitor to the toxin (*Pellegrini et al., 2005*). Zoanthids contain a potent toxin, palytoxin (*Moore & Scheuer, 1971*), and bacteria isolated from zoanthid display the presence of this hemolytic toxin as well (*Seemann et al., 2009*). The release of these toxin-forming microbes from the organism's surface, as well as the toxin directly from zoanthid, may be causing a more stressful environment and the enhancement of the toxin/antitoxin pathways in the aura-biome. The increase of *Pseudoalteromonas* and *Alteromonas* in the water column surrounding the zoanthid may be another factor enabling the already documented aggressive zoanthid species to form large monophylogenetic stands (*Suchanek & Green, 1981*; *Bastidas & Bone, 1996*; *Francini-Filho & Moura, 2010*).

The separation between aura-biomes was not absolute. Twenty liters of water was collected, with mixing occurring from the surrounding water during the sampling procedure. Despite the potential mixing of the boundary layer and surrounding water, each aura-biome showed a varying proportion of taxa and functions in the metagenomes (Fig. 6). The coral and water column metagenomes shared many genera found on coral reefs from around the world (*Ritchie, 2006*; *Wegley et al., 2007*; *Bruce et al., 2012*). The aura-biome

induced by the turf algae was driven by an abundance of *Vibrio, Flavobacterium* and *Rhodopirellula*, which is consistent with previous metagenomic descriptions of microbes present on degraded coral reefs (*Dinsdale et al., 2008a*). Zoanthids are a dominant organism on Brazilian coral reefs; these organisms are aggressive in their ability to monopolize reef space and prohibit recruitment of other species (*Mendonça-Neto & Da Gama, 2009*). The ability of zoanthid to influence the microbes in the surrounding water could be an additional invasive mechanisms.

## CONCLUSION

The exudates from the benthic reef organisms are influencing the microbial community in the water column immediately surrounding the macro-organism, creating a unique aura-biome. A combination of these aura-biomes make up the microbiome of a reef. Each aura-biome possesses functions which may drive interactions with their neighboring organisms, and some of these interactions may be negative. Therefore, as the benthic cover on a coral reef changes, the microbial community is also changing and may affect the ability of benthic organisms to recruit and grow on the reef.

### Funding

Elizabeth A. Dinsdale is funded by NSF Division of Undergraduate Education # 1323809 and Division of Microbial Biology # 1330800. Thompson, Amado-Filho, Francini-Filho, Silveira, and Moura are supported by FAPERJ, CAPES and CNPq. The funders had no role in study design, data collection and analysis, decision to publish, or preparation of the manuscript.

### Grant Disclosures

The following grant information was disclosed by the authors:
NSF Division of Undergraduate Education: # 1323809.
Division of Microbial Biology: # 1330800.
FAPERJ.
CAPES.
CNPq.

### Competing Interests

Fabiano Thompson is an Academic Editor for PeerJ.

### Author Contributions

- Kevin Walsh conceived and designed the experiments, performed the experiments, analyzed the data, wrote the paper, prepared figures and/or tables.
- J. Matthew Haggerty analyzed the data, reviewed drafts of the paper.
- Michael P. Doane, John J. Hansen, Megan M. Morris, Ana Paula B. Moreira, Louisi de Oliveira, Luciana Leomil and Gizele D. Garcia performed the experiments.

# PeerJ

- Fabiano Thompson contributed reagents/materials/analysis tools, reviewed drafts of the paper.
- Elizabeth A. Dinsdale conceived and designed the experiments, contributed reagents/materials/analysis tools, reviewed drafts of the paper.

## Field Study Permissions

The following information was supplied relating to field study approvals (i.e., approving body and any reference numbers):

The research was conducted under a federal government license (SISBIO no. 10112 - 2). We received this license to access protected areas from Parque Nacional Marinho de Abrolhos/IBAMA (Instituto Brasileiro do Meio Ambiente e dos Recursos Naturais Renova'veis).

## Data Availability

All metagenomic data described here are accessible via MG-RAST. MG-RAST ID's: Fleshy macro-algae 1-4618073.3, Fleshy macro-algae 2-4618074.3, Fleshy macro-algae 3-4618075.3, M. braziliensis-1-4618077.3, M. braziliensis-2-4618738.3, M. braziliensis-3-4618078.3, M. braziliensis-4-4618079.3, Turf algae 1-4618076.3, Turf algae 2-4618080.3, Turf algae 3-4618081.3, Water Column 1-4618082.3, Water Column 2-4618083.3, Water Column 3-4618084.3, Water Column 4-4618445.3, P. caribaeorum 1- 4618085.3, P. caribaeorum 2-4618086.3

## Supplemental Information

Supplemental information for this article can be found online at http://dx.doi.org/10.7717/peerj.3666#supplemental-information.

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
