# Peer review of "Aura-biomes are present in the water layer above coral reef benthic macro-organisms"

_PeerJ, doi:10.7717/peerj.3666_

## Round 0.1 · original submission · Major Revisions

I concur with the reviewers' comments, and ask that you undertake major revisions to improve the manuscript's writing style and structure, as well as address the reviewers' concerns.

Reviewer 1 ·

Basic reporting

The manuscript “Aura-biomes are present in the boundary layer above coral reef benthic macro-organisms” (#13683) reports on microbial communities in the water column compared to those extracted from the water closely surrounding several macro organisms (including coral, bivalves, macro-algae, turf algae, and zoanthids) in a coral reef area in Abroholes Archipelago, Brasil.
The work assesses the bacterial diversity and describes certain environmental conditions measured in the reef area. The manuscript seems to constitute an interesting contribution and provide new insights on interactions and feedback mechanisms including microbes and macrobes in the coral reef environment.

However, in the current shape there are essential data missing at large, especially but not only in the Methods section. It is neither clear how microbial diversity was assessed (e.g. primer used and PCR conditions, shotgun vs. amplicon sequencing?) nor how taxonomic classification was performed. The results lack an overview table, e.g. an equivalent to an OTU/species abundance table listing all identified microbes (are the primers specific to microbes in general or bacteria only?) and their abundance. Based on the lack of basic information, at this stage an assessment of the analysis and findings is impossible.

The authors are strongly recommend to overwork the whole manuscript in respect to author guidelines and language. In detail, please review the Peerj submission guidelines (e.g. Figure legends and Acknowledgements). Additionally, the whole manuscript but especially the Methods section doesn’t flow well. Paragraphs seem loosely connected at times or completely disconnected. Phrasing seems colloquial at times throughout the manuscript. Figures should be reviewed (e.g. Figure 2: concentration/temperature over depth seems a more intelligible way to visualize the data) A thorough editing process is suggested prior to resubmission.

Experimental design

The experimental design seems promising, but cannot be assessed at this stage due to the lack of essential data regarding the analysis of samples.

Validity of the findings

The validity of the findings cannot be assessed at this stage due to the lack of essential data regarding the analysis of samples.

Additional comments

I do encourage the authors to carefully review the manuscript as I see the value of this research.

Reviewer 2 ·

Basic reporting

Meets Standards

Experimental design

Meets Standards

Validity of the findings

no comments

Additional comments

Overall;
This study uses a metagenomics approach to profile the microbial taxa and functional pathways in the water surrounding key benthic reef taxa and compares to the bulk seawater away from these benthic species. The data generated is interesting and valuable and shows distinctive differences between the surrounding bulk water of each benthic organism. It indicates that exudates from the benthic organisms influence the microbiome. This finding is not surprising and would be expected. The functional profiling does show interesting patterns and provides some clues on how a changed microbiome may play a role in reef health and influence the fitness of surrounding organisms. The study is worthy of publication from the wealth of data and there is an interesting story here that supports previous work showing shifting reef microbial communities playing critical roles in coral reef health and potentially influencing shifts from coral to algal dominated reefs. However the manuscript falls down in many areas and often oversteps the mark of what can be interpreted from the results. In many parts highlighted below the manuscript is poorly written and it lacks clarity. While having whole community understanding by undertaking genome reconstruction would be advantageous, the approach based on profiling short reads through MG-RAST is still valuable. However the results section is quite tedious to read since it essentially is a profiling of taxa and then functional pathway hierarchies. To improve the manuscript flow and the story I would recommend combining the results and discussion sections, which will have the added bonus of reducing some repetition in the discussion. I am unsure if PeerJ accepts a combined results/discussion but think for this data set and the story it would help the manuscript immensely. At many points below it is also highlight where too much speculation is present in the manuscript or where the statements overstep the mark of what can be supported by the data. This manuscript will be a valuable contribution to the field though it needs some careful thought and revisions to bring t to a higher standard.

Individual Points:
• Abstract Lines 11-12: Untrue that most reefs around the world have transitioned from coral to algal dominated. Reefs in some locations have but there are many reefs worldwide still dominated by corals. Rephrase as this is a gross overstatement.
• Briefly provide some study details in abstract. i.e. Volume of water sampled? How were the microbial taxa and functional potential elucidated? (i.e. metagenomic approach analysed in MG-RASt)
• How were taxa and metabolic pathways influenced? i.e. summarise the important information/conclusions as the best abstract stands alone from the paper in providing a succinct overview of the paper.
• Line 25-26: No evidence or data in this manuscript regarding recruitment or growth. This is not the focus of the study so should not be the major conclusion/outcome in the abstract.
• Line 31: First sentence of Introduction and Abstract are identical. Rephrase one of these sentences.
• Line 35 – remove decline.
• Line 38-39: This sentence statement needs a reference to be supported.
• Lines 60-67: Manuscript Seymour et al MEPS 2005 could be cited here – looking at spatial dynamics of bacteria and viruses away from coral surfaces.
• Line 101: Fragmented sentence starting “Algae overgrow the coral…” Rephrase
• Lines 103-104: The logical steps between understanding microbial trends and reef restoration is not developed well. Perhaps detail the logic of how microbes can contribute to reef restoration more clearly. It is very true – but lead the reader through the thought process rather than making a large jump in logic.
• Line 120-122: Move this sentence to the acknowledgements.
• Line 124-125: Can more specific be provided on sampling. Was sampling just a one off sampling regime? Any temporal sampling? Was the time of day the same over the two days sampling? Also can something more specific than “as near as possible” be provided?
• Line 127: Remove – start sentence with “Water was collected…”
• Line 130: Obviously impossible to assess if in-situ microbes were disturbed; therefore better to rephrase as – “ care was taken to not cause excessive water movement when placing quadrats to avoid disturbing the benthic organisms boundary layers” or something similar.
• 20L is a lot of water and while containing the boundary layers – would also have bulk surrounding water? Current metagenome library preps do not need excessive amounts of DNA – hence small water volumes focused on the DBL is possible. This is mainly a comment that can be discussed (in the discussion)– as obviously the benthic water samples are different from the water away from these organisms, however finer scale boundary layer assessments are possible.
• Line 138 in n=3 for the macro-algae – three replicates for each species or 1 replicate from 3 species. Not clear here.
• Line 140-141: Again be specific “ a few metres above the reef” Can you say 3-5 metres?
• Line 145-147: Were these measurements done in bulk reef water or at the surface of the benthic organisms? This is not clear.
• Line 153: Is this 20 L passed through the 20 micron and sterivex fiters? Detail.
• Line 155-156: Are the replicate sterivex – water from 1 samples passed through multiple sterivex’s or for each replicate detailed in line 137-141 representative of 1 sterivex for each of these samples. Again clarity is required here!
• Line 174: Rephrase sentence
• Line 186: “examined”
• Lines 203-205 – Is this information important. Just present the data rather than detail the times of collection.
• Line 203-212 and Figure 2: I am unsure what this data is providing to the manuscript. The water chemistry of reefs vary drastically over daily cycles; however this data is not interpreted with the main results of the study; i.e. the microbial taxonomic and functional profiles. The water chemistry at the boundary layers of the different benthic organisms will be different and that will drive the microbial communities. This data is bulk water away from the benthos I assume, hence again it seems to have little relevance. At best the figure can be in the sup section and this text can probably be removed.
• Figures 3, 5 and 6. Minor point but the colour scheme of the symbols could be more distinctive. Maybe it is just my eyes but had trouble distinguishing red and orange at times.
• Line 337 – repeat “water column”
• Line 350: Rephrase – the concept of holobiont really cannot be applied to water (i.e. it is a specific for a host an its associated microbiome). The holobiont can influence the surrounding boundary layers.
• Line 352: Since 20L is sampled – how do you know it is the boundary layer microbiome specifically? Just need to define the boundary layer as fluidics people will have a very specific view of what boundary layer is and doubtful 20 L water is included in that view.
• Lines 360-368: Just be very clear that these observations of microbial activity, microbial oxygen consumption etc. are from previous studies. On first reading it can be interpreted that the authors are referring to results from this study. An easy fix – just start as sentence as “Previous studies have…” or something similar.
• Line 371: Did you measure hypoxia in this study? Why are results in this study showing hypoxia and aura-biomes?
• Lines 370 to 380: again this whole section is discussing previous studies and not the results from this study. The discussion should focus on the results from this study.
• Line 378-379: Where is the anaerobic conditions demonstrated in this study? Is this inferred from the sequence profiling?
• Line 384-385: Was the “growth of two heterotrophic bacteria” a result of this current study or previous studies? Not clear.
• Line 397: Needs rephrasing
• Line 400: Putative pathogenic? Plus rephrase aggressive microbiome?
• Line 405-408: There is no specific evidence of toxins being released hence rephrase; in addition highly speculative linking toxins, pathogens and dominance of this zoanthid.
• Line 410: Here is where some discussion on the size of water samples taken (20L), how representative of boundary layers and possible small scale approaches can be discussed.
• Line 419-420: Unless mistaken these values of 30-60% vibrios are based on culture work where nutrient rich media in used and biases selection towards vibrio growth. Not really appropriate to relate to disease samples and molecular profiling studies rarely show high relative abundance of vibrio sequences in disease tissues.
• Line 410-427: This whole paragraph is repeating much of what has been presented and discussed already. Hence it can be significantly reduced in length.
• Line 435: This sentence is fragmented and also quite speculative that they increase coral pathogens – as there is no evidence presented here that these retrieved sequences are actually coral pathogens in this study site.
• Line 437-438: Again this study does not look at coral recruitment processes – so a concluding sentence stating the water microbiome will make conditions unsuitable is highly speculative and not supported by any data in the study. The conclusions should focus on the actual data itself and what can be supported.

---

## Round 0.2 · Major Revisions

This paper cannot be published until the writing is dramatically improved. I assume that several of the co-authors are native speakers of English, and their efforts should be brought to bear on improving the text throughout. As it is, the revised text is filled with misspellings, sentence fragments, inappropriately capitalized words, etc. Here is an example of a nonsensical sentence beginning l.387: "The exudates and shed microbes from a reef organisms and influencing the microbial community in the water column immediately surrounding the macro-organism and a combination of these microbiomes make up the microbiome of a reef." Unless this ms. is greatly improved on revision, I will reject it.

---

## Round 0.3 · accepted · Accept

i recommend publication of this more appropriately copy-edited version, which takes into account the authors' responses to the reviewers comments.